

# Interferometric Imaging with EISCAT_3D for Fine-Scale In-Beam Incoherent Scatter Spectra Measurements

Devin Huyghebaert[1,2], Björn Gustavsson[1], Juha Vierinen[1], Andreas Kvammen[1], Matthew Zettergren[3], John Swoboda[4], Ilkka Virtanen[5], Spencer Hatch[6], and Karl M. Laundal[6]

[1]Department of Physics and Technology, UiT The Arctic University of Norway, Tromsø, Norway
[2]Department of Physics and Engineering Physics, University of Saskatchewan, Saskatoon, Canada
[3]Embry-Riddle Aeronautical University, Daytona Beach, FL, USA
[4]MIT Haystack Observatory, Westford, MA, USA
[5]Space Physics and Astronomy Research Unit, University of Oulu, Oulu, Finland
[6]Department of Physics and Technology, University of Bergen, Bergen, Norway

**Correspondence:** Devin Huyghebaert (devin.r.huyghebaert@uit.no)

**Abstract.**

The 233 MHz EISCAT_3D radar system currently under construction in northern Fennoscandia will be able to resolve ionospheric structures smaller than the transmit beam dimensions through the use of interferometric imaging. This capability is made possible by the modular design and digitisation of the 119 91-antenna panels located at the main Skibotn site. The main array consists of a cluster of 109 panels, with 10 outlier panels producing unique interferometry baselines. In the present study synthesized incoherent scatter radar signal measurements are used for interferometric imaging analysis with the EISCAT_3D system. The Geospace Environment Model of Ion-Neutral Interactions (GEMINI) model is used to simulate a Kelvin-Helmholtz instability in the cusp region at 50 m resolution to obtain plasma parameters which are then used to generate the synthetic data. The ionospheric data is forward propagated to the EISCAT_3D array, noise is added to the synthetic data, and then an inversion of the data is performed to reconstruct the incoherent scatter spectra at relatively fine scales. By using Singular Value Decomposition (SVD) with Tikhonov regularization it is possible to pre-calculate the inversion matrix for a given range and look direction, with the regularization value scaled based on the SNR. The pre-calculation of the inversion matrix can reduce computational overhead in the imaging solution. This study provides a framework for data processing of ion-line incoherent scatter radar spectra to be imaged on fine-scales. Furthermore, with more development it can be used to test experimental set-ups and to design experiments for EISCAT_3D by investigating the needed integration time for various signal-to-noise ratios, beam patterns and ionospheric conditions.



## 1 Introduction

There are many different radio systems that are able to utilize interferometry to synthesize very large apertures from individual

antennas. This provides significant control and flexibility of the beam shape and direction for antenna arrays when making use of digital signal processing techniques on the separately digitised antenna data. The use of interferometry and digital signal processing of the data from the different antenna baselines allows imaging to be performed, providing details of the origin of the radio signals in 2D or 3D space (Thompson et al., 2017). Much of the origin of radio interferometric imaging can be attributed to astronomy, where distant signals are measured with very large synchronized antenna arrays (e.g, van Haarlem

et al., 2013; Ellingson et al., 2009). Here we will focus on the utilization of radar interferometry for the measurement of signals scattered by ionospheric plasma, where the transmitted signal is a known signal source.

In radio interferometric imaging the origin of the incoming signal is commonly referred to as the 'Brightness', while the measured cross-baseline signals at the antennas are referred to as the 'Visibilities' (Thompson et al., 2017). The conversion between the Brightness and Visibility can be performed with a Fourier transform, with many assumptions including that the

signal is in the far-field (plane wave assumption). This Fourier conversion between the Brightness and Visibility space is known as the van Cittert-Zernike Theorem (Thompson et al., 2017). While the signal being in the far-field is a valid assumption for astronomical sources, in the case of the relatively close ionosphere this assumption does not necessarily hold. If this does not hold, the incoming signal cannot be considered as a plane wave and potentially the reactive components of the radiation field have to be considered. This makes the analysis considerably more complicated, and will be further discussed and investigated

when describing the data processing used in the current study.

One distinction that can cause some confusion for imaging with incoherent scatter radars (ISRs) is the difference between in-beam, or sub-beam, imaging and multi-beam imaging. In the literature the multi-beam imaging technique is commonly referred to as 'volumetric imaging' (e.g., Semeter et al., 2009; Swoboda, 2017; Stamm et al., 2023). This volumetric imaging technique provides details on greater than km scales and is restricted by the radar beam size. Conversely, in-beam imaging

provides details on sub-beam scales which can potentially resolve ionospheric features smaller than 100 m (Stamm et al., 2021).

For in-beam imaging, the most prominent ISR where it has been performed is the Jicamarca radar in Peru (e.g., Hysell and Woodman, 1997; Chau and Woodman, 2001; Urco et al., 2018). The imaging was performed for large signal to noise ratio coherent scatter targets. In-beam imaging has also been investigated for determining the power of incoherent scatter for

EISCAT_3D by Stamm et al. (2021), with very promising results for the radar. The power can be used to estimate the plasma density of the ionosphere. In this study we investigate using interferometric imaging techniques to image the full incoherent scatter spectra at sub-beam scales with the large baseline layout of the EISCAT_3D radar (McCrea et al., 2015) by creating synthetic ISR interferometric data. By imaging the full spectra it is possible to obtain the ion velocity, electron temperature, and ion temperature in addition to the plasma density.

Different inversion techniques have been investigated by previous researchers for use in radar interferometric imaging. Some of these include Capon, MaxEnt, and singular value decomposition (SVD) with regularization (e.g., Chau and Woodman, 2001;





Stamm et al., 2021). The work by Stamm et al. (2021) showed that for EISCAT_3D, SVD with regularization performed well in both computational efficiency and recreating the initial synthetic electron density profile in the ionosphere. This provides the starting point and basis for the current study, where SVD with regularization is utilized in the interferometric imaging
inversion.

The current study investigates the incoherent scatter spectra mapped to the ionosphere, rather than only a power profile, and therefore the measurement of the signal in time also has to be considered. Similar to the conversion between Brightness and Visibility space for interferometric imaging, the conversion between the frequency and lag-profile, or auto-correlation function (ACF), space for a time series can be described by a Fourier transform (e.g., Lehtinen and Huuskonen, 1996). This
is known as the Wiener-Khinchin Theorem. As will be further discussed in the analysis section, we can image each lag or frequency individually to obtain an accurate representation of the initial spectra. The combination of interferometric imaging and the Wiener-Khinchin theorem allows us to image the incoherent scatter spectra with manageable matrix sizes, rather than requiring a 3D dataset matrix inversion with the third dimension being lag/frequency.

To summarize, where we expand upon the previous work by Stamm et al. (2021) is that we use a realistic ionosphere
model for generating ionospheric plasma parameters corresponding to small scale features, we account for an approximate EISCAT_3D transmit beam in the imaging process, and we determine the ion-line ISR spectrum at each fine-scale point in the ionosphere. We also further investigate the relationship between the regularization term used in the SVD inversion and how this relates to the ratio of the signal to noise standard deviation. This study therefore provides a framework for determining the electron temperatures, ion temperatures, ion velocities, and the plasma densities at relatively fine resolutions ($< 100$ m) for the
upcoming EISCAT_3D radar.

## 2   The EISCAT_3D System Description and Layout

EISCAT_3D is a novel phased array incoherent scatter radar system currently under construction in northern Fennoscandia, with a main site located at Skibotn, Norway, and receive only sites located at Karesuvanto, Finland, and Kaiseniemi, Sweden (McCrea et al., 2015). It will operate at 233 MHz and will implement multiple advanced radar hardware and processing
systems. Some of the planned advanced capabilities of the radar include:

– A large transmit power (3.3 MW $\rightarrow$ 5 MW $\rightarrow$ 10 MW, in multiple implementation stages)

– A digital receiver phased array design

– Multi-site reception of the scattered signal for tri-vector incoherent scatter measurements

– Outlier antenna panels at the main/core site (Skibotn, Norway) to perform fine-resolution in-beam interferometry

In this study we focus on the benefit of the high transmit power, digital receiver phased array design, and outlier interferometry panels at the core site. The outlier panels at the main array provide the capability for monostatic interferometric imaging of the radar signal scattered from the ionospheric plasma. The approximate distribution of the outlier panels and resulting



interferometric baselines are presented in Figure 1. Each panel consists of 91 antennas, where there are 109 panels in the core array and 10 outlier panels placed to provide unique interferometry baselines. The distance between neighboring panels in the core of the EISCAT_3D array is $\approx 7.07$ m.

Each of the panels has a receive unit that performs a stage 1 beamforming process on the incoming voltage data from the phased array antennas in the panel. The stage 1 beamforming consists of a phase shift applied to the incoming signals at each of the antennas in a panel, where the phase shift corresponds to a certain look direction. This is followed by a summation of the signals of the antennas in the panel. The results is a beam with an approximately 10 degree beamwidth. The system is capable of forming multiple beams for each panel, but for this study we only require a single wide-beam from the stage 1 beamforming process. This then results in a single complex voltage stream of data for each panel.

Due to the complexity of implementing a multi-input multi-output (MIMO) setup and the current uncertainty surrounding when it may be able to be implemented, we focus on the single-input multi-output (SIMO) imaging capability of EISCAT_3D. The MIMO setup would consist of subsets of the EISCAT_3D array each transmitting a different pulse code. This can greatly increase the number of baselines available in the interferometry analysis, but requires high SNR signals. In the current study the array is considered to transmit the same pulse code across the array, corresponding to SIMO. For an example of potential MIMO operations with EISCAT_3D, the reader is referred to Stamm et al. (2021).

## 3 Synthetic EISCAT_3D Data Interferometric Imaging

EISCAT_3D is not yet operational, and therefore a large portion of this study is committed to generating simulated EISCAT_3D measurements that are possible to obtain within the framework of the system. To generate simulated imaging measurements, we require a block of ionospheric parameters, an incoherent scatter spectra simulator, and a scheme to add realistic noise values to idealized radar measurements. Once a synthetic EISCAT_3D data set is created, we are able to investigate the scale-sizes that will be resolvable through imaging techniques. A block diagram depicting the separation of the synthetic data generation and the interferometric imaging portions of the processing chain is provided in Figure 2.

The signal processing chain is setup to be modular and a step-by-step process. Different input ionospheric models, incoherent scatter spectra generation algorithms, and incoherent scatter spectra fitting algorithms can be interchanged relatively easily. The noise calculation and imaging technique could also be changed with relatively little effort.

To summarize, we can use the modelled plasma measurements, convert them to an incoherent scatter spectrum for each point in space, and then use these spectra to determine what the EISCAT_3D radar will measure in the interferometric baseline and lag-profile space. These are measurements that will be possible with EISCAT_3D due to the digitisation of each of the 119-antenna panels in the antenna array at Skibotn. Once the cross-baseline values are measured (or simulated in the case of the current study), we can determine the incoherent scatter spectra at relatively fine resolutions in the ionosphere. This provides not only density profiles at fine resolutions (better than 100 m resolutions at E-region altitudes as was shown by Stamm et al. (2021)), but also provides measurements of electron temperatures, ion temperatures, and ion velocities at these same scales



**Figure 1. (top figure)** The approximate locations of the 119 panels at the Skibotn site for the EISCAT_3D radar system. **(bottom figure)** The interferometric baselines mirrored about the $u = 0, v = 0$ point for the panel locations shown in the top figure, color coded for the number of repeated baselines for each point in a log base-10 scale. A baseline color corresponding to 0 therefore means there is 1 baseline with that spacing.



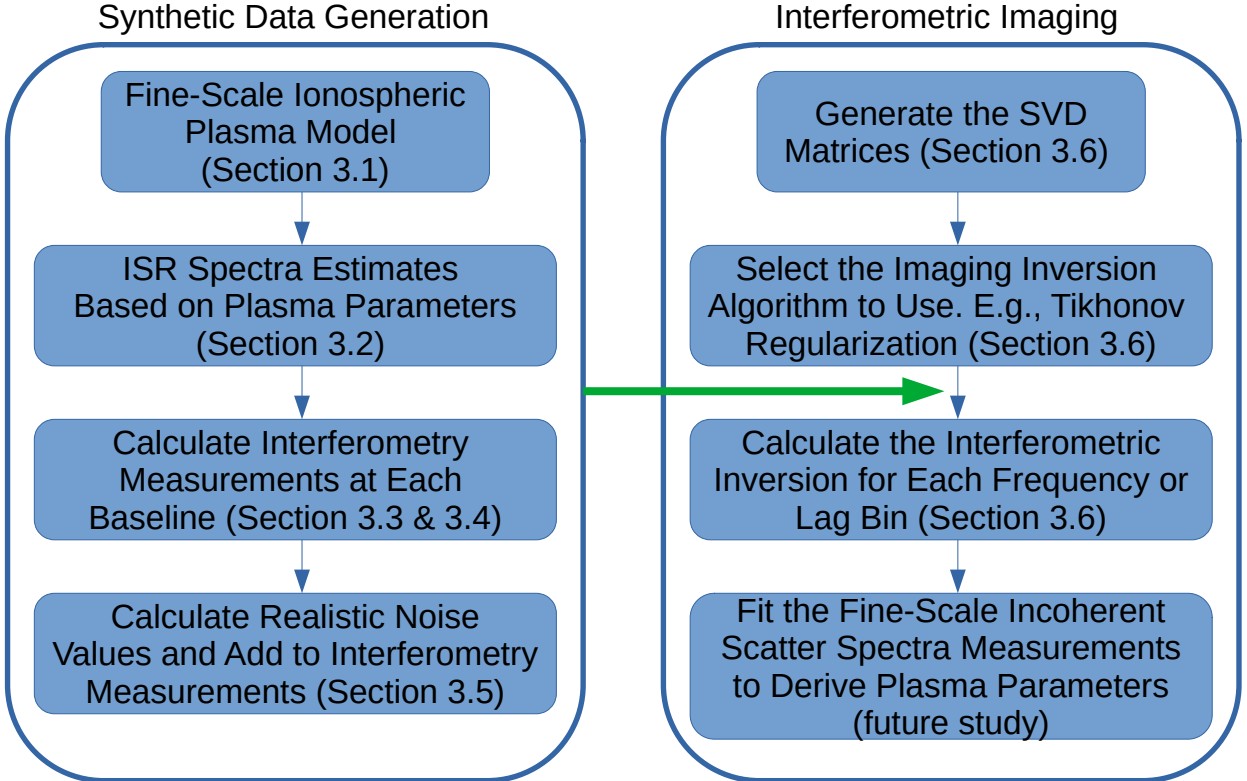

**Figure 2.** High level block diagram of the data processing chain utilized in this study. The left side shows the data processing steps used to create the synthetic EISCAT_3D dataset, while the right side displays the signal processing steps for utilization of Singular Value Decomposition (SVD) with regularization to perform interferometric imaging on the synthetic data.

115    once a fit is made to each of the spectra. There are existing packages for simulated incoherent scatter radar measurements, for example SimISR (Swoboda, 2017), but none that also include interferometric imaging.

When creating the synthetic EISCAT_3D data and performing the imaging analysis some assumptions are made. We mention these assumptions in the following list with an explanation for how they are reasonable for this analysis.

1. Plasma parameters do not vary within the range resolution of the radar

120    For the plasma parameters to not vary significantly with range, the radar would be expected to look along the geomagnetic field. This is where this interferometric imaging with EISCAT_3D will be most useful for the ionospheric parameter determination.

2. Plasma parameters are constant spatially and temporally within their 2D bin (in this study, 50 m × 50 m)

The interferometric imaging simulated here does not consider meter scale turbulence in the plasma or that the plasma

125    characteristics change over time. It is known that instabilities on meter scales exist (e.g., Fejer and Kelley, 1980; Kintner





and Seyler, 1985), but these instabilities and their associated incoherent scatter would be averaged over with the analysis presented here.

3. The 2D ionosphere is parallel with the array

In reality there will be a slight curvature of the sampled ionosphere due to the signal processing for the range determination. Due to the simulation used, we only take a 2D horizontal slice of the ionospheric parameters and consider the curvature to be small within the 5 km × 5 km area considered.

4. There is no self-clutter, or self-interference, from neighboring ranges

This is a large assumption for typical EISCAT_3D operations. Self-interference of the signal between neighboring range gates will be something that needs to be considered in the measurements. This study is a step towards interferometric imaging for general EISCAT_3D operations and we therefore ignore the self-interference effects. To include the self-interference a multi-stage processing algorithm would have to be developed. Each range in the ionosphere would be forward propagated to the antenna array, then the effects of self-interference from neighboring ranges would be included in the range being imaged, after which the imaging could proceed. This is beyond the scope of the current study.

5. All transmit power originates from the center of the core array

We consider the ionosphere to be in the far-field of the main array. The main, or core, array is where the transmitters are located and the outlier panels are receive only.

6. The side-lobe power from the transmit signal is negligible

The side lobe power of the array will be significantly reduced compared to the main lobe, and we constrain the field-of-view of the radar in the analysis.

7. There are no differences in the power received at each panel due to distance

The difference in distances for the panels of the array are at meter scales for scatter occurring at ionospheric altitudes, and are therefore not considered in the analysis of the power of the signals.

## 3.1 The GEMINI Model

The Geospace Environment Model of Ion-Neutral Interactions (GEMINI) model (Zettergren et al., 2015) was used to generate small scale ionospheric parameters for a given ionospheric instability condition. In the case of the current study, a Kelvin-Helmholtz instability was generated with a resolution of 50 m × 50 m. The initial conditions of the simulation are based on the study by Keskinen et al. (1988). The simulation had a total 2D size of 25 km × 25 km and extended from 80-1000 km in altitude with larger altitude bins as the altitude increased. In practice, any small scale fluctuations in the ionospheric plasma parameters can be input into the imaging process to examine the capability of EISCAT_3D to resolve small-scale features.

An example of the approximate EISCAT_3D array and beam size with respect to the modelled ionosphere is shown in Figure 3. Only the altitude of ≈ 250 km is shown from the model and is used in the interferometry analysis. The EISCAT_3D beam is taken to be a Gaussian shape with a full-width half-max (FWHM) of 1°, where the FWHM cone is shown in the figure.



**Figure 3.** An illustration of the approximate EISCAT_3D beam with respect to the plasma density at an altitude of 250 km altitude from the modelled ionosphere.

Figure 4 further provides the small-scale 5 km × 5 km 2D measurement box with 50 m resolution of the Kelvin-Helmholtz instability simulation at the altitude of ≈ 250 km. The size of the measurement region and the resolution were selected so that the approximate EISCAT_3D beam would be within the simulation area and so that a relatively high-end laptop (32 GB RAM, 8-core i7 2.5 GHz CPU) would be able to perform the analysis. As can be seen from Figure 3, this is only a relatively







**Figure 4.** The ionospheric plasma parameters generated from a GEMINI model Kelvin-Helmholtz instability simulation at ∼ 50 m resolution that were used in this study. The top-left figure (Ne) corresponds to the plasma density, the top-right figure (Te) corresponds to the electron temperature, the bottom-left figure (Ti) corresponds to the ion temperature, and the bottom-right figure (Vi) corresponds to the ion velocity. These data were for an altitude of ≈ 250 km.

small sample of the simulation in space, and we only take a single point in time for the ionospheric parameters. The plasma parameters are also considered to be constant within a given 3D volume element. There are plans in the future to further develop the software to consider both time variations during the measurement integration period and self-interference from neighboring

ranges when using a coded pulse.



## 3.2 Converting the Plasma Parameters to ISR Spectra

It is possible to generate an ISR spectrum for each point in 2D space using the modelled plasma parameters. We utilize the ISRSpectrum package (Swoboda et al., 2017; Swoboda, 2017) version 3.2.2. For this simulation the altitude is set to 250 km, atomic oxygen is the dominant ion, the measurements are geomagnetic field-aligned, the sample rate is 25 kHz, and the sample
time is 1 ms. These values were chosen as reasonable F-region incoherent scatter sampling experiment parameters, where the minimum range for the full ACF with 1 ms pulse length is 150 km, and the incoherent scatter spectra are within the 25 kHz bandwidth. The plasma parameters used to generate the spectra at each point are shown in Figure 4.

## 3.3 Near Field Calculations for Baselines

If we can make the far-field assumption and invoke the Wiener-Khinchin and Van-Cittert Zernike theorems, the equation for
the Visibility matrix is,

$$V(u,v,\tau) = \int \int \int B(l,m,f)\, e^{-2j\pi(ul+vm+\tau f)}\, dl\, dm\, df \tag{1}$$

where $V(u,v,\tau)$ is the matrix of complex visibility values, $u$ is the x-direction baseline in wavelengths, $v$ is the y-direction baseline in wavelengths, $\tau$ is the lag-profile spacing in time, $B(l,m,f)$ is the matrix of incoherent scatter spectra corresponding to the electron velocity distribution in the ionosphere, $l$ is the x-direction cosine, $m$ is the y-direction cosine, and $f$ is the
frequency.

To determine if we can use far-field assumptions in the interferometric imaging, we calculate the approximate Fraunhofer diffraction region for the antenna array. This provides the distance within which near-field effects need to be taken into account in the analysis of radio signals, and is given by the equation (Hysell, 2018),

$$d_F = \frac{2D^2}{\lambda} \tag{2}$$

where $D$ is the length of the baseline and $\lambda$ is the radar wavelength. For the main array of EISCAT_3D the longest baseline is approximately 80 m resulting in a Fraunhofer diffraction region of $\approx 10$ km. This means that the ionospheric scatter we observe with the main central part of the array is in the far-field. Now, if we instead consider the baselines with the outlier panels included at the Skibotn site, the longest baseline is on the order of 1 km. At these distances, the Fraunhofer diffraction region is $\approx 1500$ km. We must consider near-field effects if we include the outlier panels in the analysis. The Arecibo ISR also
had to consider near-field effects in some measurements due to the size of the radar aperture (Aponte et al., 2007).

That stated, we must also consider if the reactive components of the radiated field should also be considered. There is a boundary where there is a transition from the reactive near field region to the Fresnel, or radiative near field, region (Hysell, 2018). The approximation for the transition from the reactive to radiative regions is where the distance is greater than approximately the radar wavelength. Due to the ionosphere being 10s of km away, for ionospheric scatter we can therefore safely only
consider the radiative component of the electromagnetic field.

Based on this analysis we cannot assume plane-wave geometry when considering the outlier panels for the interferometry imaging solution, but we can consider the array to be in the radiative region. We do make some assumptions regarding the





transition from near-field to far-field in our data setup to reduce the computation requirements and the memory requirements for the inversion matrices. When we analyze the core antenna array baselines, we consider it to be reasonable to combine

repeated baselines. This is based on the 10 km Fraunhofer diffraction distance for the main array calculated earlier. Our 7021 total baselines based on the N(N+1)/2 calculation (or double this if repeated in $(u, v)$ space) can then be reduced to 2679 baselines including a mirroring of the baselines around the $(u = 0, v = 0)$ point.

To confirm that combining the repeated baselines for the main array antennas is reasonable, the distances for the baselines considering all panels were calculated. The standard deviation of the distances for each baseline in a repeated set was deter-

mined for each point in the 2D ionosphere. The mean of these standard deviation values were then taken for each baseline. The mean standard deviation in the distances is $\approx$ 1 cm, which is a small fraction of the 1.29 m wavelength. The error is therefore on the order of 1%. This is an acceptable trade-off for the reduction in memory and computational requirements that combining the core array baselines provides.

Another assumption we make is that the transmit signal originates from the center of the main array. We do not consider the

contributions from each of the panels to the transmit signal, and instead consider the transmitted signal to be a plane-wave at ionospheric altitudes. It is only for the scattered signal that we consider the effects of spherical propagation of the signal from each of the scattering regions. To summarize, we assume the transmitted signal is in the far-field, and the received signal at the different outlier arrays is in the radiative near-field.

### 3.4 Calculating the Visibility Values for Each Baseline

For the baselines, we pre-calculate the forward matrix of the system. This involves the integration of the brightness ($B$) over the $x, y, f$ space, where $x$ and $y$ are the distances along each corresponding axis in the ionosphere. We have converted from using $l, m$ to using $x, y$ as we wish to perform the imaging in the near field. To calculate the forward matrix Equation 1 is discretized and, as mentioned, we no longer consider a plane wave solution as the ionosphere is in the radiative near-field when including the outlier interferometry panels. The distance from the center of the main array to the scattering point and then to

each panel is calculated to account for the near field. We use the same forward matrix for the different frequency bins in the incoherent scatter spectra, and convert to lag-space after the interferometric transformation is performed.

The equation for the conversion of the incoherent scatter spectra sampled in the ionosphere to the Visibility measurements at the EISCAT_3D baselines is then,

$$V[u,v,\tau] = \int V[u,v,f]\, e^{-2j\pi\tau f}\, df = \int \sum_x \sum_y B[x,y,f] \left[ e^{-2j\pi(d_1[x,y,u,v] - d_2[x,y,u,v])/\lambda}\, \Delta x\, \Delta y \right] e^{-2j\pi\tau f}\, df \qquad (3)$$

where $d_1$ is the distance to one panel of a baseline, and $d_2$ is the distance to another panel. $\lambda$ is the wavelength of the radar.

The expression in the square brackets of Equation 3 can be used as a forward matrix for the imaging inversion in a linear equation. Explicitly, the forward matrix is given by,

$$A = \left[ e^{-2j\pi(d_1[x,y,u,v] - d_2[x,y,u,v])/\lambda}\, \Delta x\, \Delta y \right] \qquad (4)$$



This complex-valued matrix has dimensions [10000,2679] (x and y each have 100 points and there are 2679 baselines calcu-
lated). One might consider this problem then to be under determined. We must therefore add additional information to obtain
a sensible solution, which we do by adding Tikhonov regularization (described in Section 3.6). We can use this forward matrix
to convert the incoherent scatter spectra on the 2D grid to the Visibility $(u,v,\tau)$ space. Each frequency can be propagated in-
dividually, then converted to lag-space, lag-profiles, or auto-correlation functions (whichever term the reader prefers) once all
the frequencies are included in the Visibility value for each baseline. Effectively, from Equation 3, we first perform the imaging
for each $x,y$ point, one frequency at a time. Once this is calculated, the Fourier transform of the resulting Visibilities $(u,v,f)$
in the frequency domain are taken to convert to the lag-profile, or ACF, domain.

It should be noted that the forward matrix needs to be generated for each look direction and range to be analyzed when
performing the interferometric imaging for different experiment setups, but can be pre-calculated to reduce computational
overhead. In this study we only consider a zenith look direction and a single range of 250 km. We also assume the plasma
parameters do not vary over the range resolution of the radar and therefore the geomagnetic field for this study is considered
to be vertical.

With this calculation synthetic, noise-less interferometric EISCAT_3D data is obtained. To determine the capabilities and
limits of the system, noise must be added to the measurements at each of the baselines.

### 3.5    Noise Addition to the Signal

To add noise to the interferometric EISCAT_3D data, a normally distributed complex noise source is added to each baseline.
The standard deviation of this noise is scaled by the number of baselines with respect to the $(u,v,\tau) = (0,0,0)$ point. This means
that the outlier panel baselines have a standard deviation of noise greater by 119 times than the (0,0,0) point. 119 is the total
number of panels in the array and the (0,0,0) point corresponds to the lag-0 cross-baseline of the antennas with themselves.

Figure 5 shows the complex synthetic Visibility values for EISCAT_3D with noise included. For this example the standard
deviation of the noise was relatively small, with a signal to noise standard deviation ratio of 20 dB for the outlier panels. Based
on the previous work by Stamm et al. (2021) for imaging the E-region ionosphere, a ratio of this magnitude would require
trade-offs in range and/or temporal resolution for the experiment parameters. In this study we are observing a simulation of the
F-region, and therefore the distance will further decrease the signal to noise ratio compared to the Stamm et al. (2021) study.
For now, we consider a signal to noise standard deviation ratio of 20 dB to be an achievable scenario, and investigate both
larger and smaller signal to noise standard deviation ratios in Section 4.

The synthetic data cube is for a single range of data, where the Visibility value is considered to be averaged over the time
integration period specified for the experiment. Due to the different operating and ionospheric parameters that can affect the
signal to noise standard deviation ratio, such as range resolution, time resolution, plasma density, and beam focusing/broaden-
ing, we have focused on highlighting the capabilities for a given signal to noise standard deviation ratio. Henceforth we refer
to this ratio as SNR and leave it to future studies for optimization of the experiment setup for different time resolutions, range
resolutions, and ionospheric parameters.





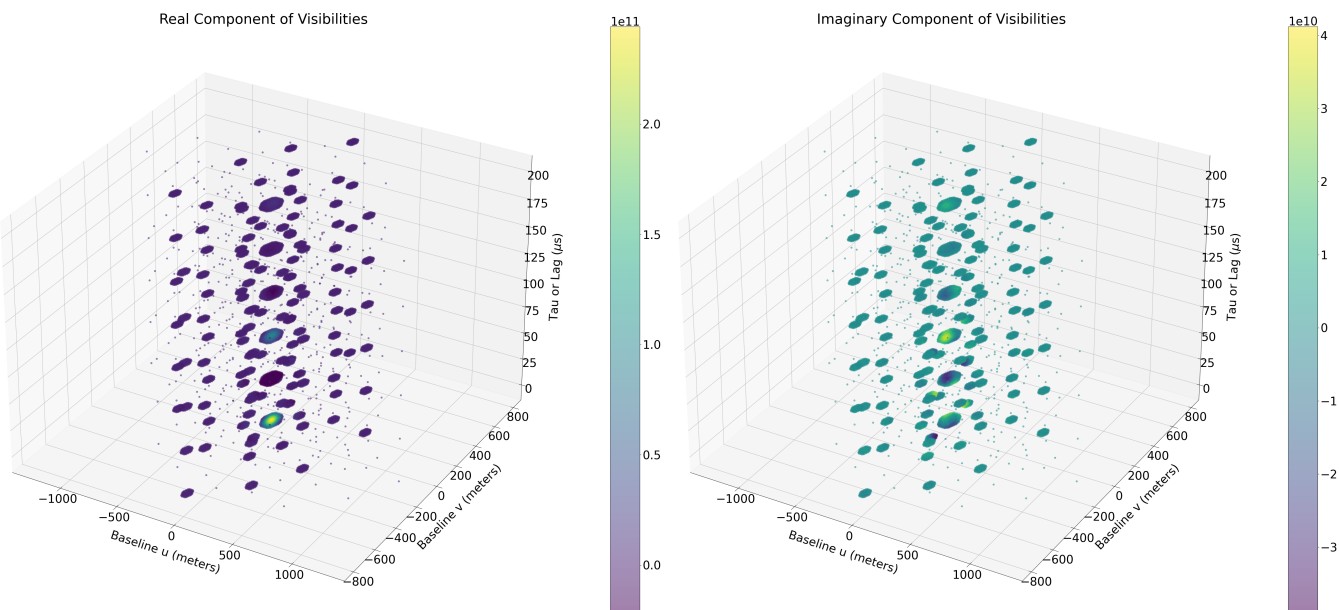

**Figure 5.** Visualization of the first lags for the 3-dimensional sparsely sampled u,v,$\tau$ space. The values of the real and imaginary components are in arbitrary units, with the added noise scaled to the values.

In this work we do not include effects caused by correlations of noise across baselines and lags. Each point in $u, v, \tau$ space is assigned a different noise value. Work has been performed previously by Hysell et al. (2008) on investigating noise correlations across lags. This can be used as a starting point for the inclusion of noise correlations in the interferometric imaging analysis for EISCAT_3D, but for now is left for future work.

### 3.6 SVD for Radar Imaging

Once we have the forward matrix and the synthetic data, we can reconstruct the initial incoherent scatter spectra on a fine-scale grid. Based on previous work by Stamm et al. (2021) we use Singular Value Decomposition (SVD) with Tikhonov regularization on the synthetic EISCAT_3D interferometric data to image the incoherent scatter spectra. The imaging can be computed relatively quickly once the initial forward matrix and SVD matrices are calculated. We are also able to scale the regularization term, and therefore the resolvable small-scale features, based on the signal to noise standard deviation ratios. The equations describing these operations follow.

We first start with a linear equation,

$$A\mathbf{x} = \mathbf{b} \tag{5}$$

where $A$ is the forward matrix described by Equation 4, $\mathbf{x}$ are the incoherent scatter spectra in the ionospheric volume sampled on a grid, and $\mathbf{b}$ are the measurements of the EISCAT_3D radar in the interferometry and frequency-space $(u, v, f)$. As



mentioned previously, this can be easily converted to the ACF domain with a Fourier transform before or after the imaging step.

We can decompose the forward matrix, $A$, using SVD. This results in the following 3 matrices,

$$A = U\Sigma V^* \tag{6}$$

where $\Sigma$ contains the singular values of $A$ arranged in descending order in a diagonal matrix, and $U$ and $V^*$ are complex unitary matrices. We then filter the singular values of $A$ with a replacement of the reciprocal of $\Sigma$ with,

$$\Sigma_{ii}^+ = \frac{\sigma_i^2}{\sigma_i^2 + \alpha^2} \frac{1}{\sigma_i} \tag{7}$$

where $\Sigma_{ii}^+$ is the new filtered diagonal singular value at matrix index $ii$, $\sigma_i$ is the original singular value for that index, and $\alpha$ is the regularization value for the 0th order Tikhonov regularization (e.g., Aster et al., 2013).

If we then calculate the inversion, we can attempt to obtain $\mathbf{x}$ from $\mathbf{b} + N$, where $N$ is the noise added to the synthetic EISCAT_3D measurements. The equation for this inversion using SVD is,

$$\mathbf{x} = V\Sigma^+ U^*(\mathbf{b} + N) \tag{8}$$

Through the use of Tikhonov regularization with SVD, we obtain the solution to,

$$\hat{\mathbf{x}}_\alpha = \mathrm{argmin}\left(||A\mathbf{x} - \mathbf{b}||^2 + \alpha||\mathbf{x}||^2\right) \tag{9}$$

which then provides an estimate of the image of one of the ISR spectral lags sampled on a 50 m × 50 m 2D grid in the ionosphere. To create the full ISR ACF this inversion is performed for each of the lags in the Visibility space (Equation 3).

An example of the results are shown in Figure 6. The left most panel is the lag-0 part of the ACF from the initial ISR spectra generated from the input plasma parameters scaled by the transmit beam radiation pattern at an altitude of 250 km, the middle panel is the imaged result using a simple summed discrete Fourier transform, while the right-most panel is the result using 0th order Tikhonov regularization with SVD. The data shown in the figure correspond to the approximate ionospheric plasma density. In practice a temperature correction needs to be applied to obtain the plasma density from these lag-0 values. This example has a relatively high signal to noise standard deviation (SNR) of 20 dB. As will be seen in Section 4, as the SNR decreases the regularization values must increase for a sensible solution. This highlights that the SNR of the signal determines the scale-size of the features that are resolvable, which is expected.

It needs to be further emphasized that each 50 m × 50 m resolution point in Figure 6 also has an ISR spectra, or electron velocity distribution, associated with it. This is highlighted in Figure 7 where the original and imaged ACFs and spectra are displayed for the point at y=0 m and x=-500 m. This, again, corresponds to a SNR of $\approx$ 20 dB and a regularization value of 20. It is clear that a reasonable representation of the initial spectrum is obtained with the imaging.

The spectra or ACFs can be fit to an incoherent scatter spectra model, providing multiple parameters of the plasma, such as the electron density, electron temperature, ion temperature, and the ion velocity. The full fit of the plasma parameters is not performed in the current study. In the future the spectra on the refined scales will be fit and a fully circular synthetic data



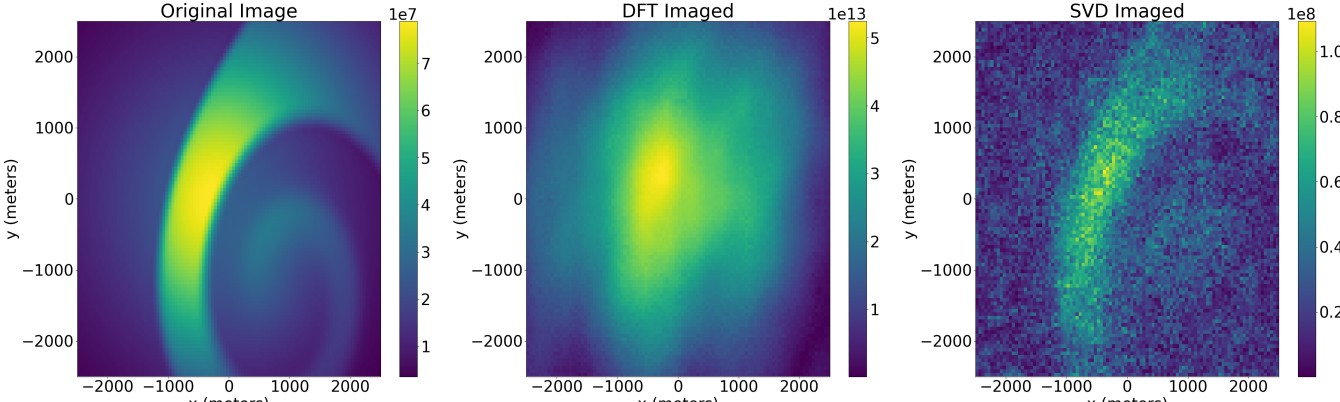

**Figure 6.** The initial $\tau$=0 power **(left figure)**, as well as the image with a generic discrete Fourier transform **(middle figure)** and imaging using 0th order Tikhonov regularization with SVD **(right figure)**. The regularization value was 20, and the SNR from the full transmit power to 2 receive panels was $\approx$ 20 dB. The values are not scaled to radar power and cross-section values, and are therefore meant as only an illustration for this study.

processing chain will be provided to the community. This will be useful for experiment design, and essential for the validation of results once EISCAT_3D is operational. The next section investigates and presents how varying the regularization parameter
and SNR affects the imaging results.

## 4 Results and Discussion

One of the benefits of utilizing SVD with Tikhonov regularization for imaging with EISCAT_3D is that it is possible to pre-calculate the inversion matrix of $V\Sigma^{+}U^{*}$ in Equation 8 for a given range and look direction, and also select a regularization value based on the SNR of the signal. This optimizes the resolvable scale sizes with the interferometric imaging and minimizes
the computation time required. The results in this section further highlight how different SNRs and regularization values can affect the results.

The time for the creation of the synthetic EISCAT_3D data for the 25 sampled frequencies ($\pm$ 12 kHz at 1 kHz resolution) was 13 s, the SVD matrix generation was 151 s, and the data inversion was 1.5 s. This was implemented with minimal parallelization on a high-end laptop (32 GB RAM, 8-core i7 2.5 GHz CPU), and therefore these times can be greatly reduced.
No GPU acceleration was implemented, which would further improve the processing times. The SVD matrix creation, which requires the largest portion of the computation time by a significant margin, can be pre-generated for a given range, resolution, and pointing direction of the radar. The Tikhonov regularization can then be implemented on these pre-generated matrices before the inversion. Through parallelization of the code, current rapid advances in parallel computing architectures, and by taking a limited window in the inversion processes, it is expected that the processing in this study will be achievable at real-time.





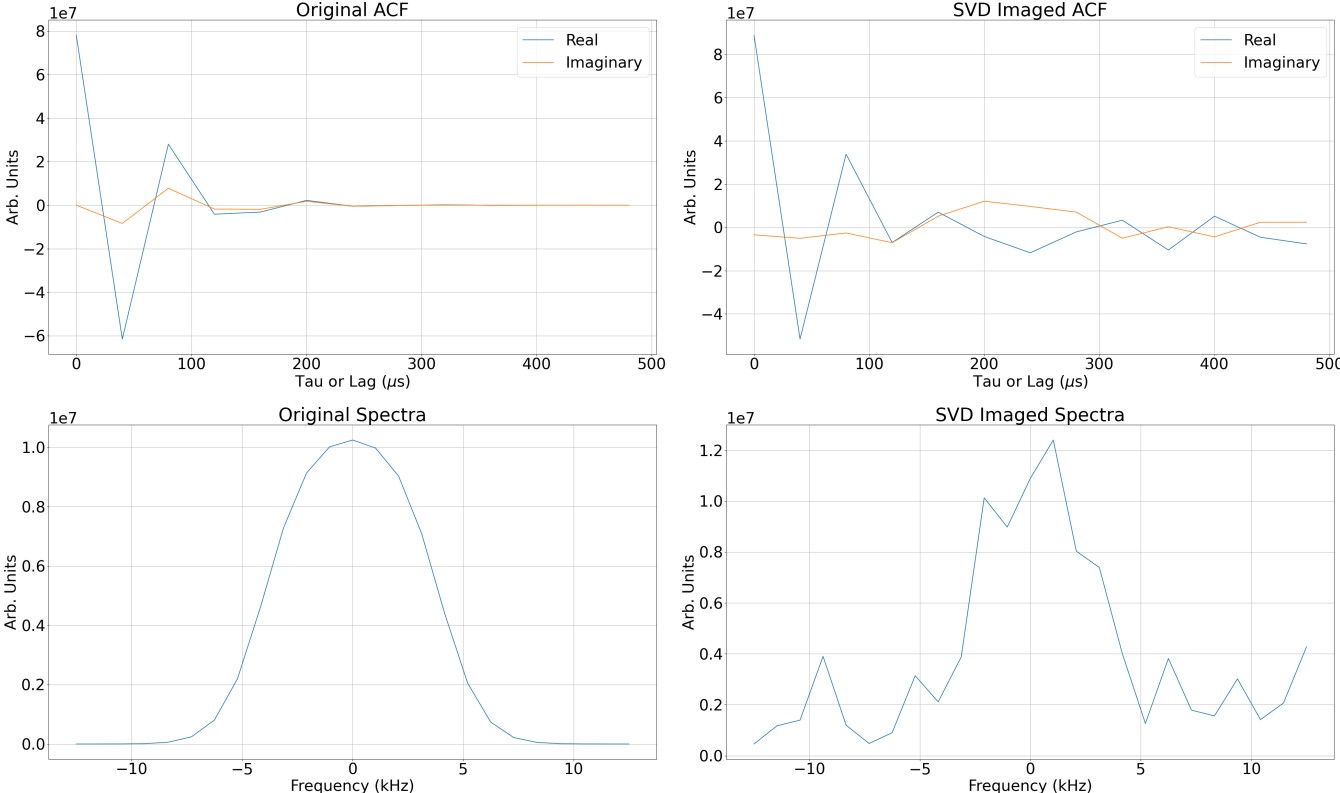

**Figure 7.** These figures represent the ACFs and corresponding spectra for the (y=0 m, x=-500 m) point from Figure 6. Each point in the 2D ionosphere plane has a spectra associated with it.

Figure 8 shows the imaged brightness values in the ionosphere for a slice through the $\tau$=0 points. This effectively provides the power profile for each point, which is related to the plasma density and temperature. The top left corner corresponds to the largest SNR and lowest regularization, with decreasing SNR moving to the right along the columns and increasing regularization moving down along the rows. The left-most column is a very large SNR for ISR measurements, and is expected to be rarely achieved in high temporal resolution with EISCAT_3D. As the SNR depends on many factors, including the

experiment parameters (range resolution, pulse length, temporal resolution, etc.), sky and system noise, and the ionospheric conditions, we only provide results based on the bulk SNR values for the current study.

    This highlights how care needs to be taken to select a suitable regularization value given the SNR of the data. If too little regularization is imposed, the imaged data is relatively unusable. If too much regularization is imposed, many of the small-scale features we are most interested in examining with the in-beam imaging are smoothed and lost in the filtering. Selecting the

optimal regularization based on the SNR is therefore an essential factor when implementing imaging with EISCAT_3D.

    One of the key places this study expands upon the previous work by Stamm et al. (2021) is the calculation of incoherent scatter spectra for each of the points. This is highlighted in Figure 9, with the panels displayed corresponding to the $y = 0$ data





**Figure 8.** Lag-0 ACF power for varying SNR (columns) and regularization (rows). The SNR refers to the ratio of the signal to standard deviation of the noise. The columns correspond to SNRs of ≈ 917, 92, 18, and 9 (≈ 30 dB, 20 dB, 13 dB, and 10 dB) from left to right. The rows correspond to regularization values of 4, 10, 20, 40, 100, and 200 from top to bottom.

slice. The regularization and SNR values are the same as Figure 8. From this we can see that incoherent scatter spectra are also resolvable depending on the regularization used and SNR of the measurements.



**Figure 9.** Same as Figure 8, but for a slice of the data along y=0 to display the frequency domain.

Figure 8 and 9 therefore provide two 2D cuts of the 3D data set and show how the imaging is affected by the SNR of the signal and the regularization parameter used. We also investigate the least-squares error of the imaged solution vs the original incoherent scatter distribution for different SNR and regularization parameters. This is shown in Figure 10. Unsurprisingly, there are optimal regularization values for a given SNR of data. There are different algorithms by which to select this regularization value for a dataset, many of which are iterative approaches that are computationally demanding. The current study is



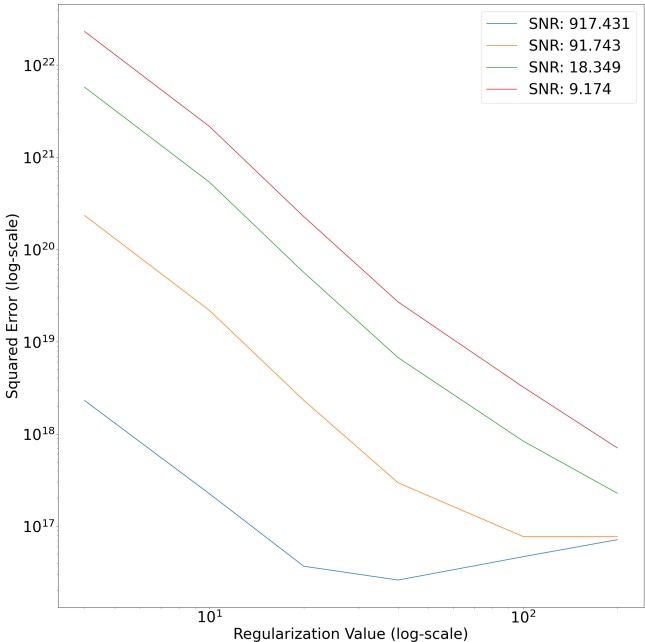

**Figure 10.** The sum of the squared errors between the initial and imaged spectra are displayed for varying regularization (x-axis) and SNR (lines) values.

meant as a case study of using regularized SVD for interferometric imaging of incoherent scatter spectra, where the optimal selection of regularization values will be investigated in a subsequent study.

## 4.1   Plasma Parameter Error Analysis

Based on the variance of the imaged ACFs, it is possible to estimate the errors of the plasma parameters from the expected ISR spectra fits. To calculate the variance, each pixel has the imaged ACF values subtracted from the original ACFs. The variance

of the resulting residual across the lags is determined. The ratio of the lag-0 power to the variance of the ACF is then calculated for each pixel.

We use the work by Vallinkoski (1988) to estimate the errors that would result from fitting the ISR spectra based on this ratio. An example of the error estimates for the different points is provided in Figure 11. A four parameter fit is assumed, where the four parameters are plasma density, ion temperature, ion-to-electron temperature ratio, and the ion velocity. The figure

corresponds to the same data as shown in Figure 6. Note again that this is for a 20 dB SNR between the transmit signal and 2 interferometer panels, and a Tikhonov regularization value of 20.

Clearly with the imaging setup and SNR used in this study, we should be able to resolve the incoherent scatter spectra and resulting ionospheric plasma parameters in the high density regions of the radar scattering volume. The pattern of points with reduced errors are due to the pattern of panels in the EISCAT_3D array.



**Figure 11.** Estimated plasma parameter errors calculated based on the variance of the imaged ACFs. This corresponds to an SNR of $\sim 92$, and a regularization of 20. The errors are with respect to the true value for three of the four parameters, where Ne is the plasma density, Ti is the ion temperature, Tr is the ratio of ion temperature to electron temperature. Vi is the ion velocity and the error is given in m/s.

## 5 Conclusions

One of the significant breakthroughs with EISCAT_3D will be the opportunity to use interferometric imaging to measure the ionospheric plasma properties with sub-km resolution. This study further develops a method first proposed by Stamm et al. (2021) for imaging the EISCAT_3D measurements. It has been expanded to include imaging of the full ion-line spectra rather than only the power corresponding to the plasma density. The imaging of the ion-line allows fine-scale reconstructions of



multiple plasma parameters. Plasma densities, electron temperatures, ion temperatures, and ion velocities will all be able to be obtained on fine resolutions, assuming there is sufficient signal to noise standard deviation ratio (SNR). A benefit of using Tikhonov regularization with singular value decomposition (SVD) is that the inversion matrices can be pre-calculated and the regularization can be scaled based on the SNR of the signal. This results in an efficient use of computational resources. For active auroral conditions with high plasma density regions spatial resolutions on the order of 100 m or less should be

achievable.

Consistent sub-km measurements of the ionosphere are difficult to achieve with most systems. Satellite and rocket in-situ measurements can achieve these resolutions, but these instruments quickly pass through the region of interest. Other instruments that probe the ionosphere with radio waves, such as ionosondes and coherent scatter radars, typically do not achieve sub-km resolution. This can be either due to ambiguities caused by the refraction of the radar signal, or due to not

transmitting a signal with a large enough bandwidth for sub-km range resolution. Even when these other radar instruments do achieve sub-km spatial resolution, they do not provide the details about the ionospheric plasma that incoherent scatter radars are capable of. The ability to consistently measure the small-scale plasma characteristics within the transmit beam illumination region will contribute to solving many of the outstanding small-scale ionosphere questions.

We have showcased the benefit of coupling a fine-scale ionospheric electrodynamics model to the processing chain, along

with the utilization of existing ISR spectra generation software in the processing. These tools will be used to characterize observability of various types of small-scale structures with EISCAT_3D in the future. This can be expanded upon from the Kelvin-Helmholtz simulation presented here.

The software is setup to be modular, so the software packages for the data processing stages in this study can be replaced. With a synthetic interferometric dataset for EISCAT_3D it is possible to start developing processing algorithms that can be

implemented on EISCAT_3D computing servers by the EISCAT association, mitigating security risks associated with users accessing the radar hardware.

While this study focuses on interferometric imaging of the ionospheric plasma ion-line in fine scale resolution, the imaging technique is not limited to only this use. It is possible to also image, for example, polar mesospheric summer echoes (PMSE), sporadic E-layers, plasma lines, and naturally enhanced ion-acoustic lines on sub-km scales. Some modifications to the initial

data setup and experiment design may be required depending on the spectra of the phenomena, but the signal processing concepts are the same. The imaging techniques presented here can be also combined with volumetric multi-beam techniques, providing details about ionospheric features on both small and large scales.

Future work will include a fully integrated programming stack that takes the user defined radar experiment parameters, including the time and range resolution, determines the expected signal to noise standard deviation ratio, selects an appropriate

regularization value from the expected SNR, then performs the imaging. A full 3D in-beam profile can then be created for the different parameters, with each range sampled from a fine-scale ionospheric dynamics model (GEMINI in the case of the current study) and processed separately. The software will be made publicly available once the full in-beam profile for multiple ranges is implemented. A future addition will include self-interference effects from neighboring range gates.



*Code availability.*  The GEMINI3D ionospheric model code is open source and, along with documentation, can be obtained from
( https://github.com/gemini3d/ ). The ISRSpectrum code and documentation can be obtained from
( https://github.com/jswoboda/ISRSpectrum ).

*Author contributions.*  DH, BG, and JV conceptualized the study. DH and BG created the interferometric imaging software. DH, BG, JV, and
IV debugged the imaging software. AK and MZ generated the fine-scale ionospheric parameters using the GEMINI model. JS contributed
the software to obtain the ISR spectra from the plasma parameters. IV contributed the error analysis from the imaged spectra. SH and KML
contributed to the matrix inversion descriptions and how best to describe the results. All authors contributed to the writing, reviewing, and
editing of the manuscript.

*Competing interests.*  The authors declare no competing interests.

*Acknowledgements.*  The basis for this work was presented by Huyghebaert et al. (2023) at URSI GASS 2023 in Sapporo, Japan. DH
is funded through a UiT The Arctic University of Norway contribution to the EISCAT 3D project funded by Research Council of Norway
through research infrastructure grant 245683. AK is funded by the Research Council of Norway through the CASCADE project through grant
326039. IV is supported by the Research Council of Finland project 347796. SMH and KML were funded by the Trond Mohn Foundation
and by the Research Council of Norway, project 300844/F50. Discussions regarding this research were supported by the International Space
Science Institute (ISSI) in Bern, through ISSI International Team project #506 (Understanding Mesoscale Ionospheric Electrodynamics Using
Regional Data Assimilation). Discussions with Andres Spicher on investigating ionospheric turbulence using EISCAT_3D interferometric
imaging are acknowledged.



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
