# Peer review of "Simulation of Interferometric Imaging with EISCAT\_3D for Fine-Scale In-Beam Incoherent Scatter Spectra Measurements"

_EGUsphere, 2024_

## Referee Comment (RC1)

The manuscript presents really interesting simulation results and clearly demonstrates the efficiency of constructed EISCAT_3D facility in fine-structure detection and resolution using in-beam imaging. The authors performed a thorough analysis of simulated incoherent scatter spectra employing the model of ionospheric parameters and synthesized noise to retrieve the ionospheric signature of Kelvin-Helmholtz instability. The authors used well-proven theoretical methods for data analysis and provided in-depth description and discussion. As a whole, the manuscript contains many important outputs that are useful for many scientists and other stakeholders. The results obtained can be developed further and will be useful for testing future experimental results and arranging new promising experiments. I hope, the EISCAT_3D facility will be put into operation very soon and provide new insights into the high-latitude ionospheric plasma.

The manuscript is worth to be published after some revisions. My specific comments are below.

1. As for me, the title of the manuscript reflects the main idea slightly incorrectly. The reader can wrongly assume that the authors dealt with real experimental data. I suggest adding something like "synthetized EISCAT_3D data", "simulated imaging", "Theoretisation and simulation" or something else – it is up to the authors.

2. Line 169: "atomic oxygen is the dominant ion…". This is unclear, if the authors consider pure atomic oxygen ions (oxygen approximation)? If not, what other heavy or light ion fractions they analyzed?

3. Line 317: "…data for the 25 sampled frequencies (± 12 kHz at 1 kHz resolution)…". As for me, the used spectrum is sampled really roughly. I might be wrong but can suppose that it can result in the distortions both the incoherent scatter signal and the autocorrelation function. I had some experience in incoherent scatter signal synthesis (but for multiple ion species), where the sampling of the order of 10 Hz should be applied. I realize that the laptop productivity is limited. It would be good to refer some publications or justify adequacy of the used resolution.

4. If I am right, the authors also simulated the noise to add it to the incoherent scatter signal. Often, the real noise characteristics are far from the synthesized ones. It would be better, if the authors used the characteristics of really measured noise in the same location (of course, if possible).

5. I have noticed that the manuscript suffers from the lack of quantitative estimations, while the authors prefer the qualitative assessments. For instance, "Selecting the optimal regularization based on the SNR is therefore an essential factor…" (lines 334 – 335), "…incoherent scatter spectra are also resolvable depending on the regularization used and SNR of the measurements" (page 17, bottom), "…assuming there is sufficient signal to noise standard deviation ratio (SNR)" (line 366). Fortunately, Figures 8 – 10 can shed some light but the reader need to guess what values the authors meant as "optimal" or "sufficient". Taking into account, that the reported method is supposed to be

used further, it would be useful to emphasize on the specific values, where this method is appropriate. I suggest adding the table with SNR and regularization coefficients (maybe, something else) to quantitatively specify the results.

6. Line 371: "Consistent sub-km measurements…". I think that this paragraph should be omitted or moved to Introduction. This shifts the focus from the results to the advantages of the EISCAT_3D facility on the whole.

---

## Referee Comment (RC2)

This manuscript demonstrates via numerical simulation the in-beam interferometric imaging observations of ionospheric fine-scale structures using the EISCAT_3D radar, which is currently under construction in northern Fennoscandia and will be operational in the near future. The authors created ionospheric parameters by simulating the Kelvin-Helmholtz instability with the GEMINI model and converted them to incoherent scatter radar spectra with the ISRSpectrum package. The simulated radar data were then created by calculating interferometry measurements and adding normally distributed noise to them. The inversion was performed by Singular Value Decomposition (SVD) with Tikhonov regularization. This manuscript is useful to readers because it carefully describes the background of in-beam interferometric imaging observations, including many assumptions, and evaluates the performance of this method quantitatively. In addition, the authors provide information on computation time and consider future real-time data processing, which is important for the actual radar observations. Therefore, I believe this manuscript will be accepted for publication in Annales Geophysicae after some revisions.

**Moderate comments:**

1. Subsection 3.3:   As far as I understand this subsection, the signal transmitted from the main panels is the far-field and the signal received by the outlier arrays is the radiative near-field. However, I think the description in this subsection may be somewhat confusing to the readers. Thus, I suggest that the authors explain at the beginning of the subsection the three types of electromagnetic fields, i.e., the far-field, radiative near-field, and reactive near-field, as well as a summary of this subsection.

2. In my opinion, the optimal value of the regularization parameter ($\alpha$) cannot be

determined from SNR only, but should be tuned according to the type, spatial structure, and spatial scale of target phenomenon. If that is correct, it is difficult to predetermine $\alpha$ and time-consuming to determine the optimal value of $\alpha$ dynamically. Do authors have any ideas on how to determine it?

**Minor comments:**

1. Figure 1: The authors should explain what $u$ and $v$ mean in the caption.

2. Line 201: What does "N" mean? Is it the number of antenna panels?

   If so, is it N(N-1)/2 instead of N(N+1)/2?

3. Line 225: What are $d_1$ and $d_2$ the distances from? From the scattering point (x,y)?

4. Line 290: I think that $\alpha$ in Equation (9) is a mistake for $\alpha^2$, or $\alpha^2$ in Equation (7) is a mistake for $\alpha$. Please check them.

5. Line 327: "moving to the right along the columns". → along the rows?

6. Line 328: "moving down along the rows". → along the columns?

7. Line 388-389 (after "for example"): I recommend that the authors add several reference papers for these various ionospheric phenomena.

---

## Author Comment (AC1)

Dear Reviewers and Editor,

Thank you for the reviews on the manuscript. The comments and suggestions are appreciated, with responses to each provided below.

Sincerely, The Authors

Reviewer 1:

The manuscript presents really interesting simulation results and clearly demonstrates the efficiency of constructed EISCAT\_3D facility in fine-structure detection and resolution using inbeam imaging. The authors performed a thorough analysis of simulated incoherent scatter spectra employing the model of ionospheric parameters and synthesized noise to retrieve the ionospheric signature of Kelvin-Helmholtz instability. The authors used well-proven theoretical methods for data analysis and provided in-depth description and discussion. As a whole, the manuscript contains many important outputs that are useful for many scientists and other stakeholders. The results obtained can be developed further and will be useful for testing future experimental results and arranging new promising experiments. I hope, the EISCAT\_3D facility will be put into operation very soon and provide new insights into the high-latitude ionospheric plasma.

The manuscript is worth to be published after some revisions. My specific comments are below.

1. As for me, the title of the manuscript reflects the main idea slightly incorrectly. The reader can wrongly assume that the authors dealt with real experimental data. I suggest adding something like "synthetized EISCAT\_3D data", "simulated imaging", "Theoretisation and simulation" or something else – it is up to the authors.

**Reply:**

The new title is now:

'Simulation of Interferometric Imaging with EISCAT\_3D for Fine-Scale In-Beam Incoherent Scatter Spectra Measurements'

2. Line 169: "atomic oxygen is the dominant ion…". This is unclear, if the authors consider pure atomic oxygen ions (oxygen approximation)? If not, what other heavy or light ion fractions they analyzed?

**Reply:**

Atomic oxygen is the only ion in the spectra calculation - this has been clarified in the text.

3. Line 317: "...data for the 25 sampled frequencies (± 12 kHz at 1 kHz resolution)...". As for me, the used spectrum is sampled really roughly. I might be wrong but can suppose that it can result in the distortions both the incoherent scatter signal and the autocorrelation function. I had some experience in incoherent scatter signal synthesis (but for multiple ion species), where the sampling of the order of 10 Hz should be applied. I realize that the laptop productivity is limited. It would be good to refer some publications or justify adequacy of the used resolution.

**Reply:**

For F-region incoherent scatter spectra, this resolution will prove sufficient for a fit of the spectra. To obtain 10 Hz resolution spectra, inter-pulse lags are required to be calculated. Expanding the simulation to include user input parameters for pulse length and sampling rate is underway. As shown in Figure 7, the current simulation

setup provides a relatively good sampling of the ISR spectrum for the altitude considered. The simulation provides sufficient details on the viability of using the interferometric imaging capability of EISCAT\_3D for investigating the ion-line in high spatial resolution. Further improvements to the software for parallelization of the imaging process for each lag, or tau, will allow much higher frequency resolution analysis – though this is constrained by the pulse-length unless inter-pulse lags are included in the simulation.

4. If I am right, the authors also simulated the noise to add it to the incoherent scatter signal. Often, the real noise characteristics are far from the synthesized ones. It would be better, if the authors used the characteristics of really measured noise in the same location (of course, if possible).

**Reply:**

Ideally we would have real noise, and there are plans to improve the simulated noise characteristics to better match reality in the future. There are many assumptions that go into using 'ideal' noise sources, such as that the noise is not correlated across panels or time. For now, we do not have access to real noise characteristics from an EISCAT\_3D panel and have to rely on simulations.

5. I have noticed that the manuscript suffers from the lack of quantitative estimations, while the authors prefer the qualitative assessments. For instance, "Selecting the optimal regularization based on the SNR is therefore an essential factor..." (lines 334 – 335), "…incoherent scatter spectra are also resolvable depending on the regularization used and SNR of the measurements" (page 17, bottom), "…assuming there is sufficient signal to noise standard deviation ratio (SNR)" (line 366). Fortunately, Figures 8 – 10 can shed some light but the reader need to guess what values the authors meant as "optimal" or "sufficient". Taking into account, that the reported method is supposed to be used further, it would be useful to emphasize on the specific values, where this method is appropriate. I suggest adding the table with SNR and regularization coefficients (maybe, something else) to quantitatively specify the results.

**Reply:**

There are many factors that will go into determining what is considered 'optimal' resolution and acceptable plasma parameter value errors for a given experiment. As also stated by Reviewer 2, the determination of the 'optimal' regularization value is a difficult task. What we are attempting to highlight in this manuscript is that the signal-to-standard deviation ratio greatly affects what scales of features are able to be resolved with interferometric imaging with EISCAT\_3D. Some experimenters may sacrifice spatial resolution for time resolution for rapidly evolving ionospheric phenomena. There are plans for future studies to investigate these very things. The least squares errors is provided in Figure 10, but this does not provide the full picture of resolution trade-offs for different signal-to-noise standard deviation ratios and regularization values as you mention.

A comparison of the plasma parameters input to the simulation with plasma parameters output from the simulation is the best way forward. An incoherent scatter spectra fitting routine is planned to be included in the future which will then provide these comparisons.

6. Line 371: "Consistent sub-km measurements...". I think that this paragraph should be omitted or moved to Introduction. This shifts the focus from the results to the advantages of the EISCAT\_3D facility on the whole.

**Reply:**

The paragraph has been moved to the introduction.

**Reviewer 2:**

This manuscript demonstrates via numerical simulation the in-beam interferometric imaging observations of ionospheric fine-scale structures using the EISCAT\_3D radar, which is currently under construction in northern Fennoscandia and will be operational in the near future. The authors created ionospheric parameters by simulating the Kelvin-Helmholtz instability with the GEMINI model and converted them to incoherent scatter radar spectra with the ISRSpectrum package. The simulated radar data were then created by calculating interferometry measurements and adding normally distributed noise to them. The inversion was performed by Singular Value Decomposition (SVD) with Tikhonov regularization. This manuscript is useful to readers because it carefully describes the background of in-beam interferometric imaging observations, including many assumptions, and evaluates the performance of this method quantitatively. In addition, the authors provide information on computation time and consider future real-time data processing, which is important for the actual radar observations. Therefore, I believe this manuscript will be accepted for publication in Annales Geophysicae after some revisions.

Moderate comments:

1. Subsection 3.3:

As far as I understand this subsection, the signal transmitted from the main panels is the far-field and the signal received by the outlier arrays is the radiative near-field. However, I think the description in this subsection may be somewhat confusing to the readers. Thus, I suggest that the authors explain at the beginning of the subsection the three types of electromagnetic fields, i.e., the far-field, radiative near-field, and reactive near-field, as well as a summary of this subsection.

**Reply:**

The description of the radiative and reactive near-field and the radiative far-field have been further described in the section.

2. In my opinion, the optimal value of the regularization parameter ( $\alpha$ ) cannot be determined from SNR only, but should be tuned according to the type, spatial structure, and spatial scale of target phenomenon. If that is correct, it is difficult to predetermine  $\alpha$  and time-consuming to determine the optimal value of  $\alpha$  dynamically. Do authors have any ideas on how to determine it?

**Reply:**

The authors agree that determining the regularization parameter is a difficult endeavor. There are also many different regularization methods that could be further investigated.

Future studies are planned to simulate EISCAT\_3D imaging with different ionospheric phenomena generated with the GEMINI model. This can help with determining what regularization values are optimal for a given event. The simulated imaging analysis of different phenomena coupled with optical imager data during actual operations could greatly constrain the optimal regulation value space that needs to be searched, making iterative algorithms potentially feasible.

A short description of different regularization value selection methods is now added in the Results and Discussion section.

Minor comments:

1. Figure 1: The authors should explain what u and v mean in the caption. **Reply:** The description has been added.

2. Line 201: What does "N" mean? Is it the number of antenna panels? If so, is it N(N-1)/2 instead of N(N+1)/2? **Reply:** Correct. Thank you for catching the mistake.

3. Line 225: What are d 1 and d 2 the distances from? From the scattering point (x,y)? **Reply:** The distances are from the center of the array to the scattering point added with the distance of the scattering point to a panel. This has been clarified in the text.

4. Line 290: I think that  $\alpha$  in Equation (9) is a mistake for  $\alpha$  2, or  $\alpha$  2 in Equation (7) is a mistake for  $\alpha$ . Please check them. **Reply:** Equation 9 should have alpha^2. Thank you for catching this mistake.

5. Line 327: "moving to the right along the columns". along the rows? **Reply:** This has been clarified.

6. Line 328: "moving down along the rows". along the columns? **Reply:** This has been clarified.

7. Line 388-389 (after "for example"): I recommend that the authors add several reference papers for these various ionospheric phenomena. **Reply:** Reference papers have been added for the different ionospheric phenomena.

---

## Referee Report (RR1)

The manuscript presents important simulation results and clearly demonstrates the efficiency of constructed EISCAT_3D facility in fine-structure detection and resolution using in-beam imaging. The authors performed a thorough analysis of simulated incoherent scatter spectra employing the model of ionospheric parameters and synthesized noise to retrieve the ionospheric signature of Kelvin-Helmholtz instability. The authors used well-proven theoretical methods for data analysis and provided in-depth description and discussion. As a whole, the manuscript contains many important outputs that are useful for many scientists and other stakeholders. The results obtained can be developed further and will be useful for testing future experimental results and arranging new promising experiments.

The authors made all revisions suggested by me, additional clarifications and explanations. I think that after this revision, the manuscript is worth to be published as is.